# Decawave UWB Clock Drift Correction and Power Self-Calibration

**DOI:** 10.3390/s19132942

**Published:** 2019-07-04

**Authors:** Juri Sidorenko, Volker Schatz, Norbert Scherer-Negenborn, Michael Arens, Urs Hugentobler

**Affiliations:** 1Fraunhofer Institute of Optronics, System Technologies and Image Exploitation IOSB, 76275 Ettlingen, Germany; 2Institute of Astronomical and Physical Geodesy, Technical University of Munich, 80333 Munich, Germany

**Keywords:** ultra-wideband (UWB), time of arrival (TOA), navigation

## Abstract

The position accuracy based on Decawave Ultra-Wideband (UWB) is affected mainly by three factors: hardware delays, clock drift, and signal power. This article discusses the last two factors. The general approach to clock drift correction uses the phase-locked loop (PLL) integrator, which we show is subject to signal power variations, and therefore, is less suitable for clock drift correction. The general approach to the estimation of signal power correction curves requires additional measurement equipment. This article presents a new method for obtaining the curve without additional hardware and clock drift correction without the PLL integrator. Both correction methods were fused together to improve two-way ranging (TWR).

## 1. Introduction

In the last century, autonomous systems became omnipresent in almost every field of the industry. One of the most important tasks in robotics is the interaction between a robot and its environment. This task can only be accomplished if the location of the robot with respect to its environment is known. Visual sensors are very common for localization [1,2]. In some cases, estimating the position in non-line-of-sight conditions is required. Radio-frequency-based (RF) sensors are able to operate in such conditions, but the outcome depends highly on measurement principles, such as received signal strength indicator (RSSI) [3], fingerprinting [4], FMCW [5] and UWB [6], as well as on techniques such as the angle of arrival [7], time of arrival [8] or time difference of arrival [9]. Indoor positioning is, in general, a challenge for RF-based localization systems. Reflections could cause interference with the main signal. In contrast to narrowband signals are ultra-wideband (UWB) signals, which are more robust to fading [10,11]. A common UWB system is the Decawave UWB transceiver [12], which is low cost and provides centimeter precision. The accuracy and precision of this chip are affected by three factors: hardware delays, clock drift, and signal power [13,14]. This article discusses clock drift correction and signal power error, which is specific to the Decawave UWB transceiver and affects the accuracy of the position significantly. The general approach to estimating signal power dependency is to use ground truth data, which are provided by additional measurement equipment [15]. The clock drift error is caused by the different frequencies of the transceiver clocks. The general approach to Decawave UWB clock drift correction is to use the integrator of the phase-locked loop (PLL) [16,17,18,19]. In the following section, we explain that the general approach to clock drift correction is not suitable because the PLL is also affected by the signal power. Therefore, a more accurate method for clock drift correction is presented. The middle sections of this article discuss the estimation of the signal power correction curve without the need for additional hardware. As far as we know, nobody has obtained a signal power correction curve by self-calibration before. The last part of this article presents a two-way ranging (TWR) method that is able to use the correction methods for distance estimation.

## 2. Decawave UWB

Decawave transceivers are based on UWB technology and are compliant with IEEE802.15.4-2011 standards [20]. They support six frequency bands with center frequencies from 3.5 GHz to 6.5 GHz and data rates of up to 6.8 Mb/s. The bandwidth varies with the selected center frequencies from 500 up to 1000 MHz. With higher bandwidth, the send impulse becomes shorter. The timestamps for the positioning are provided by an estimation of the channel impulse response, which is obtained by correlating a known preamble sequence against the received signal and accumulating the result over a period of time. In contrast to narrowband signals, UWB is more resistant to multipath fading. Reflections would cause an additional peak in the impulse response. The probability that two peaks interfere with each other is small. The sampling of the signal is performed by an internal 64 GHz chip with 15 ps event-timing precision (4.496 mm). Because of general regulations, the transmit power density is limited to −41.3 dBm/MHz. These regulations are due to the high bandwidth occupied by the UWB transceiver. The following experiments were carried out with the Decawave EVK1000. This board mainly consists of a DW1000 chip and an STM32 ARM processor.

## 3. Clock Drift Correction

In practice, it is not possible to manufacture exactly the same clock generators, so every transceiver has a different clock frequency. Clock drift correction represents the difference between clock frequencies but not current time values.

### 3.1. General Approach

The general approach to clock drift correction is to use the PLL integrator [16,17,18,19]. Figure 1 shows an example of frequency demodulation by a PLL. The voltage-controlled oscillator (VCO) is set to the mid-position and the loop is locked in at the frequency of the carrier wave. Modulations on the carrier would cause the VCO frequency to follow the incoming signal, so changes in the voltage correspond to the applied modulations. The difference between the received carrier frequency (VE) and the internal loop frequency (VI) can be observed in the integrator of the loop filter. In Figure 2, the integrator output is presented. The test scenario is based on measurements obtained at every 50 ms between two stationary transceivers. The difference between the two frequencies is about five parts per million. Reaching the final condition took up to 15 min. The tests were repeated four times with another two stationary stations. Figure 3 shows the filtered results of the obtained curves provided by a 500-point moving average filter. The curve progression is deterministic. Decawave indicates that the logarithmic increase of the integrator at the beginning is due to the warm-up when the crystal oscillator is activated, graphically represented in Figure 4. This oscillator follows from the combination of a quartz crystal and the circuitry within the DW1000-based design.

In the following test scenario, the effect of the signal power on the integrator was investigated. Both the transmitter and receiver stations were stationary. Figure 5 shows the measured signal strength at the receiving station. After about 2340 s, we arranged the transmitter to reduce the signal power. The integrator of the receiver jumped after the signal power changed to a new level (Figure 6), indicating that distance changes between the transmitter and receiver would affect the integrator, and so, affect the clock drift correction as well.

The reason for this dependency could be the analog phase detectors of the PLL, in which the loop gain $K_{D}$ is a function of amplitude, which affects the error signal $v_{e}\left(t\right)=K_{D}\left[\Phi_{Out}\left(t\right)-\Phi_{In}\left(t\right)\right]$, and so, affects the pull-in time (total time taken by the PLL to lock) as well.

### 3.2. Proposed Approach for the Clock Drift Correction

In this section, we present an alternative method for the clock drift correction, which is independent of the signal power. The measurement setup is presented in Figure 7. All measurements and calibrations were conducted with Decawave EVK1000 boards. The station with the identification number (id) 2 is the transmitting station (TX). The receiving station (RX) has the identification number 1. The receiving signal power, as well as the timestamps, were obtained by reading the register provided by the transceivers [14,15]. The general settings for the hardware setup can be found in Table 1 and the notations in Table 2.

Figure 8 shows a schematic diagram of the approach. TX is sending three signals at times $T_{1}$, $T_{2}$, and $T_{3}$. The clocks of the transmitter and receiver are not synchronous. If the clocks have no drift, then both clocks should have the same frequency and the difference between $\Delta T_{1,2}=T_{2}-T_{1}$ should be the same for the transmitter and the receiver; otherwise, $\Delta T_{1,2}^{RX}\neq \Delta T_{1,2}^{TX}$. The same applies to $\Delta T_{1,3}$. If the clock of RX is running faster than that of TX, then $\Delta T_{1,3}^{RX}>\Delta T_{1,3}^{TX}$ and the clock drift error becomes $C_{1,2}=\Delta T_{1,2}^{RX}-\Delta T_{1,2}^{TX}$.

Previously, the frequency difference between the two clocks was presented by the integrator of the PLL. After the warm-up time, the clocks reached their final frequencies. The clock error now increased linearly. For short measurement periods the clock drift error can be assumed to be linear even during the the oscillator’s warm-up.

The main idea is for the clock drift error $C_{1,3}=\Delta T_{1,3}^{RX}-\Delta T_{1,3}^{TX}$ to be used for correcting the timestamp $T_{2}$ by simple linear interpolation. In Figure 9, three messages, P1, P2, and P3, with constant signal powers have been sent. The delay between every message was about 2 ms. The values are already filtered; hence, every point consists of the mean of 2000 measurements. The Figure 10 shows the clock drift error $C_{1,2}=\Delta T_{1,2}^{RX}-\Delta T_{1,2}^{TX}$. Because of the long delay, the distance error resulting from the clock drift is about 1 m.

In the next step, the clock drift error $C_{1,2}$ is corrected by the linear interpolation of $C_{1,3}$.

(1)$$C_{1,2}^{{}^{\prime }}=C_{1,2}-\frac{C_{1,3}}{\Delta T_{1,3}^{TX}}\cdot \Delta T_{1,2}^{TX}$$

The results are shown in Figure 11. The correction requires only three messages and the remaining average offset is about $-1.915\;\cdot \;10^{-5}$ m. The linear interpolation is also suitable for the warm-up phase. The implementation of the presented clock drift correction for the TWR is presented in the last section. A position error caused by a constant velocity of the object is also corrected by the linear interpolation, due to the linear increase of the position error (pseudo clock drift). In practice, it is possible to obtain $\Delta T_{1,3}^{TX}$ = 1 ms. An acceleration high enough to cause an error greater than 5 mm would require almost 1000 g $\left(10^{4}\frac{m}{s^{2}}\right)$.

## 4. Signal Power Correction

The next section discusses the signal power correction. It is known that the time stamp of the DW1000 is affected by the signal power, in which an increase causes a negative shift of the time stamp and vice versa.

### 4.1. General Approach

Figure 12 illustrates the reported distance error with respect to the received signal power. At a certain signal strength, the range bias effect should be zero. In Figure 12 the bias vanishes between −80 and −75 dBm. The correction curve is affected by the system design elements, such as printed circuit boards, antenna gain, and pulse repetition frequency (PRF). The general approach to correction curve estimation is to compare the distance measurements with the ground truth distances. This method has two disadvantages. First of all, additional measurement equipment is necessary. Second, every created curve applies to two stations but not every individual station.

Figure 13 shows the relationship between the measured and correct signal strengths for different PRF. The measured signal power is correct only for measurements smaller than −85 dBm. The knowledge of the difference between the measured and correct signal strengths can be used for additional measurement techniques, such as the RSSI, for distance estimation.

### 4.2. Proposed Approach for the Signal Power Correction

In the previous section, we discussed an alternative approach to clock drift correction with three messages (P1, P2, and P3). The following method is based on this concept, but the TX station changes the signal strength of the second message (P2). Figure 14 shows how the signal strengths of the first and last messages (P1 and P3) remain constant and only the signal strength of the second signal (P2) decreases after 1000 measurements. Every measurement point is the result of the mean of 2000 signals. The tests were conducted with a cable connection of 10 cm and the transmitter decreased the signal gain with a step size of 3 dB. The transmit power settings can be adjusted by changing the gain of the transmit Driver Amplifier (DA) and the transmit Mixer. These changes are not equivalent to the output power. Figure 11 shows that, after the clock drift correction, the remaining error of $C_{1,2}^{{}^{\prime }}$ (Equation 1) is close to zero. With the decreasing signal strength of P2, the error of $C_{1,2}$ is increasing, see Figure 15. Hence, it is possible to create a dependency between the measured signal strength and the timestamp error.

In the following test scenario, the power calibration was repeated with an antenna and a distance of 1.5m between the RX and TX stations. The gain step size was reduced to 0.5dB. Figure 16 and Figure 17 shows the results of the filtered signal power calibration curve. The main difference between Decawave’s curve, as shown in Figure 12, and our curve is that the zero line is unknown. This line marks the signal power at which the timestamp error is zero. The step size of the decreasing transmitting signal power gain was constant, but the measured decreasing signal power curve for P2 was nonlinear because the measured signal power did not equate to the correct signal power for high signal strength, as shown in Figure 12.

It is necessary to pay attention to the timing between the messages. With short delays between the messages, it is possible that they affect each other. This effect can be seen by the offset between P1 and P3 in Figure 18. In Figure 16 a delay of 2 ms was used between the messages and in Figure 18 a delay of 150 μs was used.

It was previously mentioned that the measured signal strength equals the correct signal power only for small signal powers. Therefore, it is possible to use the very first measurements with small signal strengths to estimate the slope. Figure 19 shows an estimated line based on the estimated slope. The results are the same as the curve obtained by Decawave except that no additional measurement equipment is required and our curves can be obtained individually for every station. Figure 20 illustrates the correction curve with respect to the signal power.

Even for the same hardware design, it is possible that the shape of the correction curve differs. In Figure 21 and Figure 22, the final results of the power correction curve are obtained from another station. The calibration was repeated six times. The shapes of the curves are deterministic but different from those of the station above. Therefore, it makes sense to repeat the calibration for every individual station.

## 5. Two Way Ranging

The following section describes how the presented clock drift and signal power correction can be used for precise TWR. Figure 23 shows the concept for the TWR. The initial message is sent by the reference station at $T_{1}^{R}$ and received by the tag. The timestamp $T_{2}^{T}$ is affected by the signal power and causes an error E1. After some delay caused by internal processing, the tag sends a response message at $T_{2}^{T}$. The reference station receives the response from the tag and saves the timestamp $T_{2}^{R}$, which is affected by the signal power error E2. In this example, the delay due to the hardware offset is not considered.

The time of flight between the reference station and the tag can be determined by the following formula. It is assumed that the distance between the two devices does not change between time stamp $T_{2}^{R}$ and $T_{1}^{R}$.

(2)$$T_{TOA}=\frac{\left(T_{2}^{R}-T_{1}^{R}\right)-\left(T_{2}^{T}-T_{1}^{T}\right)-E_{2}-E_{1}}{2}$$

The values E1 and E2 can be obtained from the signal power correction curve. It should be taken into account that the signal power affects the tag and reference station differently. The time difference $\Delta T_{1,2}^{R}$ increases with decreasing signal power. The zero lines for both the signal power and hardware offset are unknown but constant; hence, both values are represented by the variable Z. In the previous section, we explained that the clock drift could be corrected by three messages. Figure 24 shows how this principle can be adapted for TWR. The last message was used to obtain the clock drift error $C_{1,3}=\Delta T_{1,3}^{R}-\Delta T_{1,3}^{T}$. The signal power E1 had no effect on the time stamp difference $\Delta T_{1,3}^{T}$. The final time of the flight equation with the clock drift correction becomes: $$\begin{array}{c} T_{TOA}=0.5\cdot \left(\Delta T_{1,2}^{R}-\Delta T_{1,2}^{T}-\frac{C_{1,3}^{RT}}{\Delta T_{1,3}^{T}}\cdot \left(\Delta T_{1,2}^{T}+E1\right)\right)+0.5\cdot \left(-E_{2}-E_{1}\right)+Z \end{array}$$

The results of the TWR with signal power and clock drift correction are illustrated in Figure 25. The blue line represents the difference between laser distance measurements (ground truth) and distances provided by the TWR. The 11 distances extend from 3.515m to 0.562m. Every point results from the mean of 2000 measurements. The unknown hardware offset, which causes the 0.3m offset, is not relevant in this example. The signal power error depends on the distance and the clock drift on time. If both effects are corrected properly, the resulting difference between the mean error and every measurement error should be as small as possible. The standard deviation of the error is 0.015 m. The small error difference between the blue and red line shows that the signal power and clock drift correction are both sufficient. The antenna area was 0.0012m${}^{2}$; therefore, it is not possible to obtain ground truth data with a precision higher than a few centimeters.

## 6. Conclusions

This article presents a new method for signal power and clock drift correction. It was shown that the curves obtained for the signal power correction could be highly accurate and deterministic, as well as provide individual results for every station. The signal power correction procedure can be performed once as a factory calibration. In addition to the estimation of the signal power correction curve, it was also possible to obtain the relationship between the measured and real signal powers. Knowing the relationship allows for better distance estimations with methods based on the signal strength. In contrast to the general approach, our clock drift correction is independent of the signal power and promises results with centimeter accuracy. The last part of the article explained how the signal power and clock drift correction are fused together to provide highly accurate TWR.

## 7. Patents

Patent pending. 

## Figures and Tables

**Figure 1 sensors-19-02942-f001:**
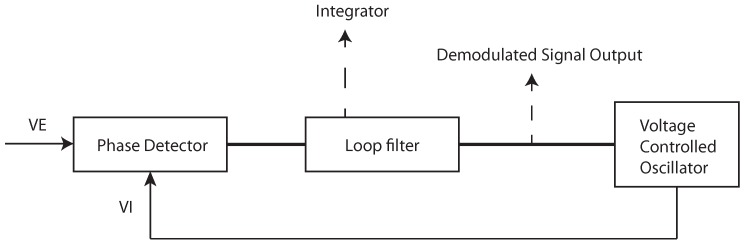
Example of the phase locked loop (PLL).

**Figure 2 sensors-19-02942-f002:**
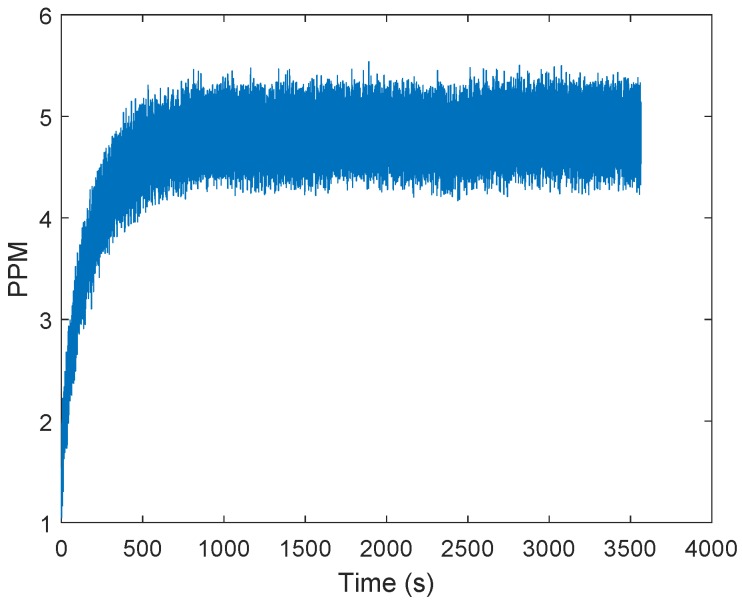
Frequency difference between the received carrier and the internal phase locked loop (PLL) in parts per million (PPM). The curve is obtained by reading the carrier integrator value of the DW1000 chips. The logarithmic increase of the curve is due to the warm-up of the crystal oscillator.

**Figure 3 sensors-19-02942-f003:**
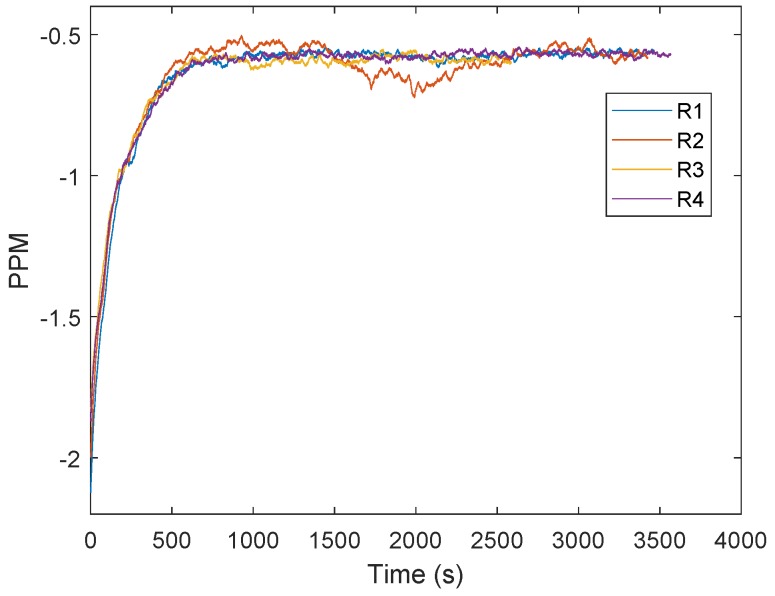
Filtered frequency difference between the received carrier and the internal phase locked loop (PLL) in parts per million (PPM). The colors represent different measurement obtained successively. It can be seen that the curves are deterministic.

**Figure 4 sensors-19-02942-f004:**
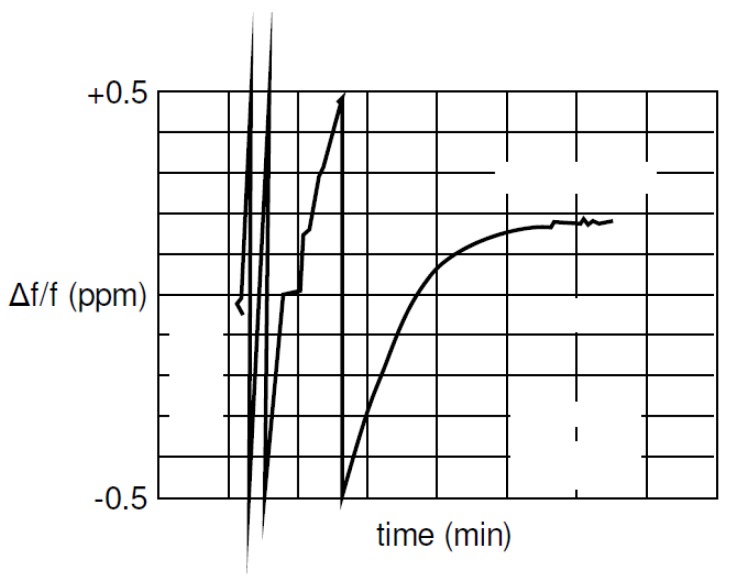
Frequency difference during the warm-up phase of the DW1000 crystal oscillator in parts per million (PPM) with respect to the time in minutes [15]. The figure is used with permission [15].

**Figure 5 sensors-19-02942-f005:**
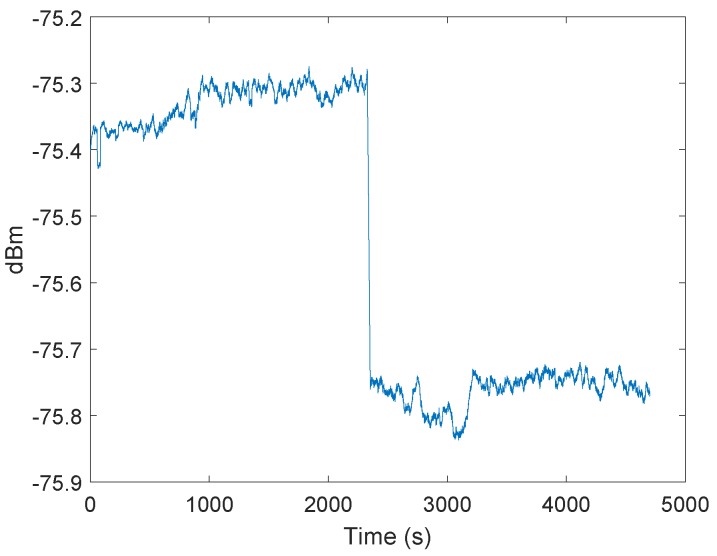
Signal power in dBm of the received blink message. The curve shows the filtered results of the received signal power over time, measured by the DW1000 chip. After 4600 measurements the transmitting signal power was reduced.

**Figure 6 sensors-19-02942-f006:**
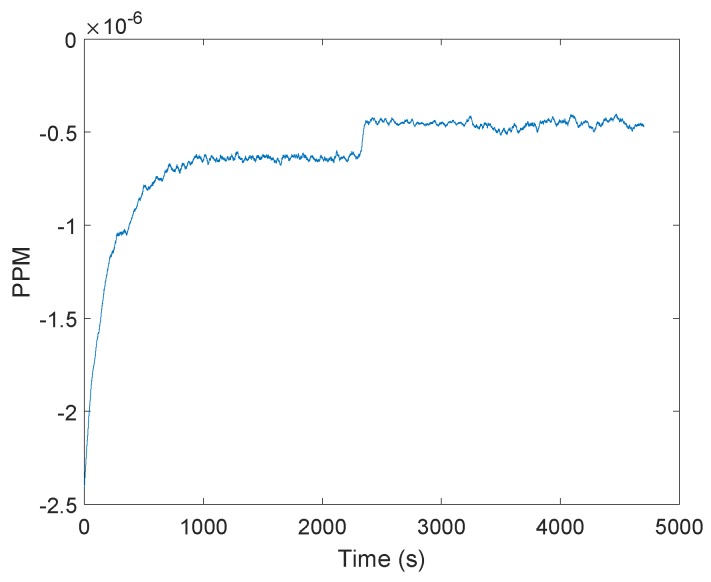
Filtered frequency differences between the received carrier and the internal phase locked loop (PLL) in parts per million (PPM) after changing the transmitting signal power after about 2340 s.

**Figure 7 sensors-19-02942-f007:**
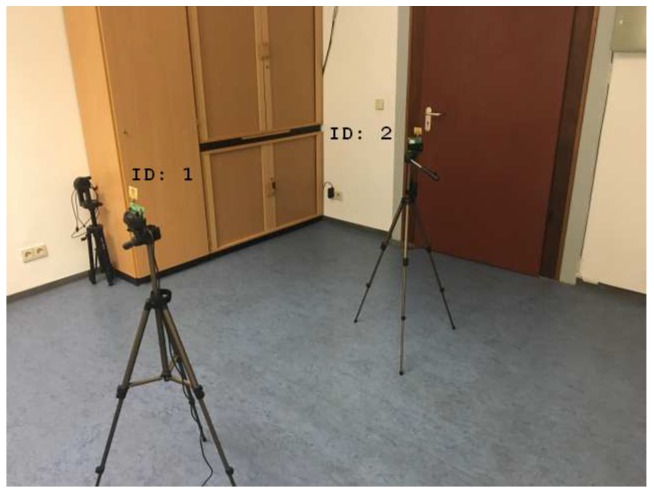
The measurement setup consist of two transceivers (EVK100) with the identification number 1 and 2.

**Figure 8 sensors-19-02942-f008:**
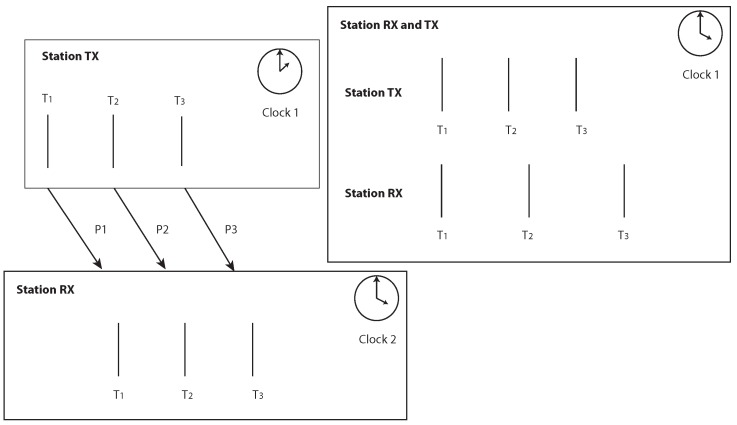
Schematic for the presented clock drift correction based on three transmitting messages. See text for explanation.

**Figure 9 sensors-19-02942-f009:**
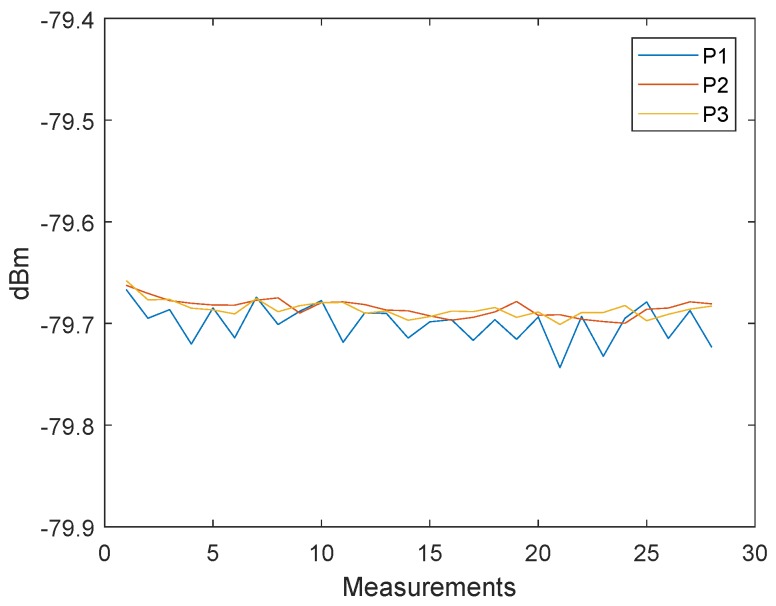
Filtered received signal power measurements of the three messages: P1, P2 and P3.

**Figure 10 sensors-19-02942-f010:**
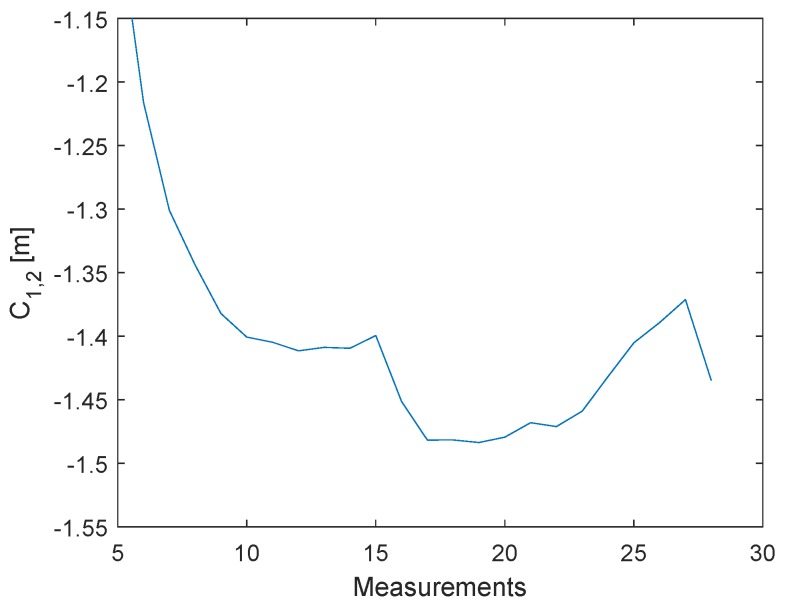
The error in meters caused by the clock drift. The curve is changing, due to the clock warm-up.

**Figure 11 sensors-19-02942-f011:**
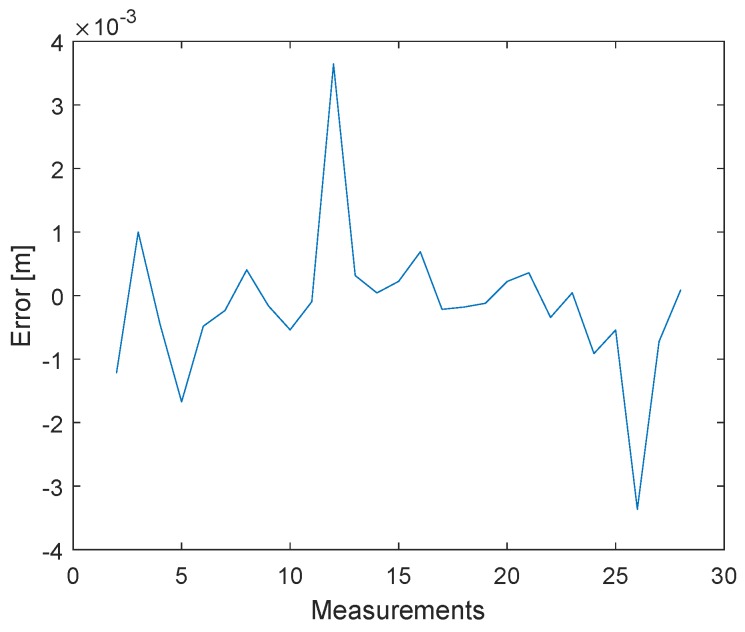
The error in meters after the clock drift correction. $C_{1,2}^{{}^{\prime }}$.

**Figure 12 sensors-19-02942-f012:**
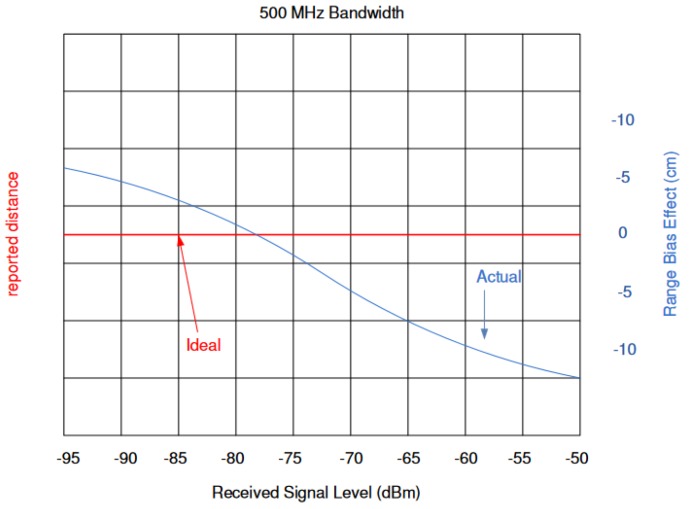
The effect of the received signal power on the distance measurement [15], used with permission. The red line represents the correct distance measurements. The blue line shows the range bias caused by different signal powers [15].

**Figure 13 sensors-19-02942-f013:**
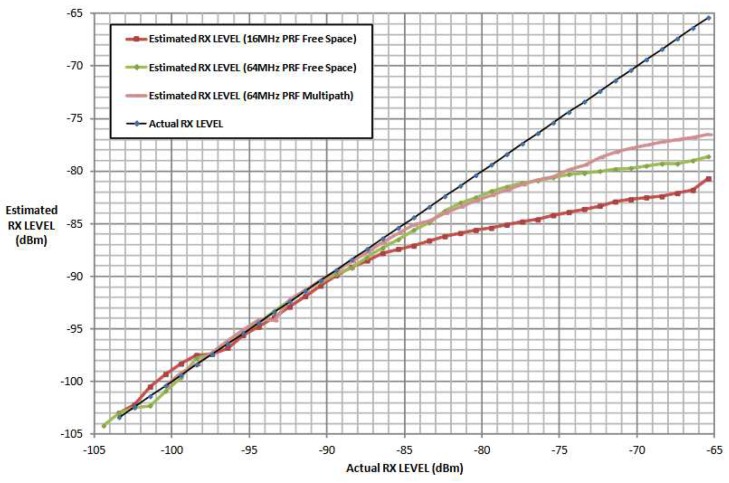
Measured received signal power at the DW1000 chip, with respect to the correct signal power. The blue line represent the reference line, every value of the x-axis has the same value on the y-axis. The other lines are the results of the estimated received signal power with different settings. The estimated received signal power equates the correct signal power for low signal strength [14].

**Figure 14 sensors-19-02942-f014:**
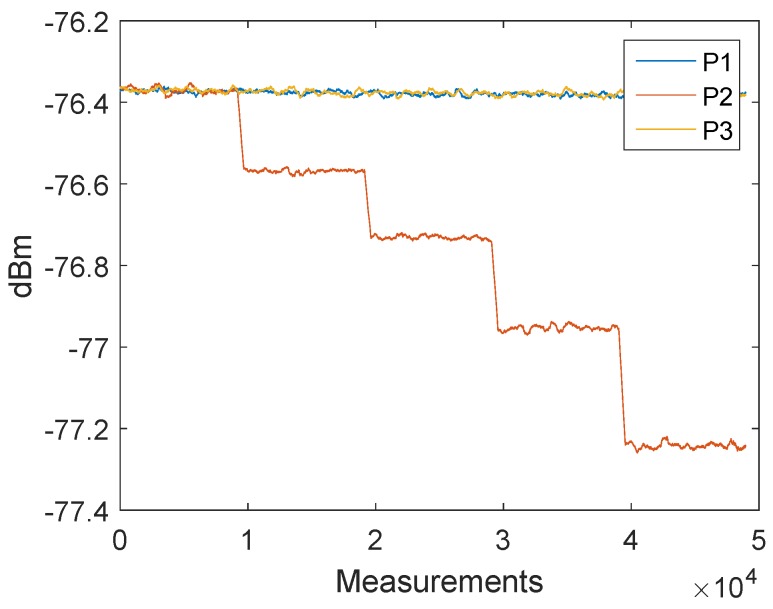
Filtered received signal power measurements of the three messages P1, P2 and P3. The measurements have been obtained with a cable connection between the transceivers. The transmitted signal power of the second signal P2 is reduced with a step size of 3 dB, while the signal power for P1 and P3 remains constant.

**Figure 15 sensors-19-02942-f015:**
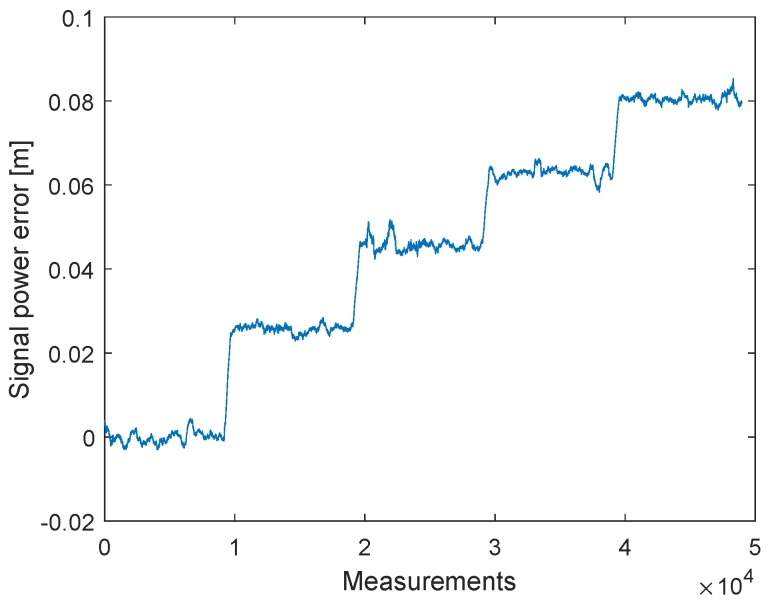
Filtered timestamp error changes due to the received signal power. The measurements have been obtained with a cable connection between the transceivers. The error is changing systematically with decreasing signal power.

**Figure 16 sensors-19-02942-f016:**
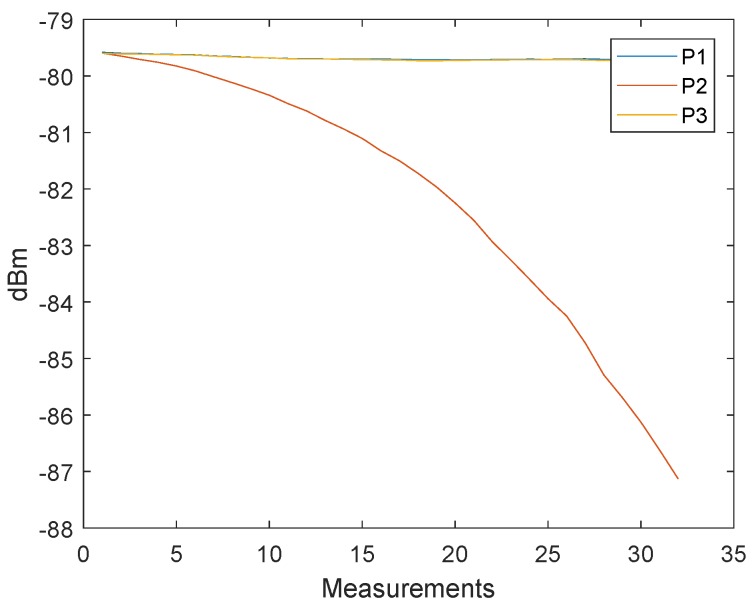
Filtered received signal power measurements of the three messages P1, P2 and P3. The measurements have been obtained with a wireless connection between the transceivers. The transmitted signal power of the second signal P2 is reduced with a step size of 0.5 dB, while the signal power for P1 and P3 remains constant.

**Figure 17 sensors-19-02942-f017:**
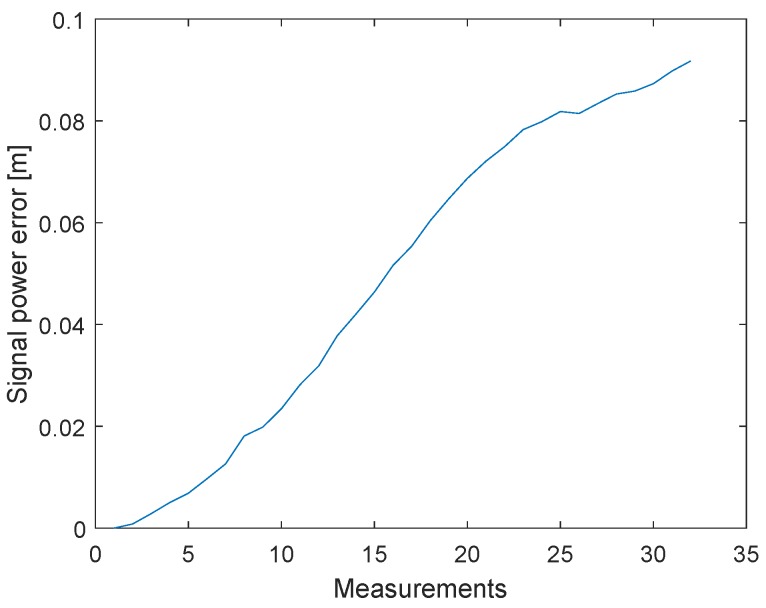
Filtered timestamp error changes due to the received signal power. The measurements have been obtained with a wireless connection between the transceivers. The error is changing with decreasing signal power.

**Figure 18 sensors-19-02942-f018:**
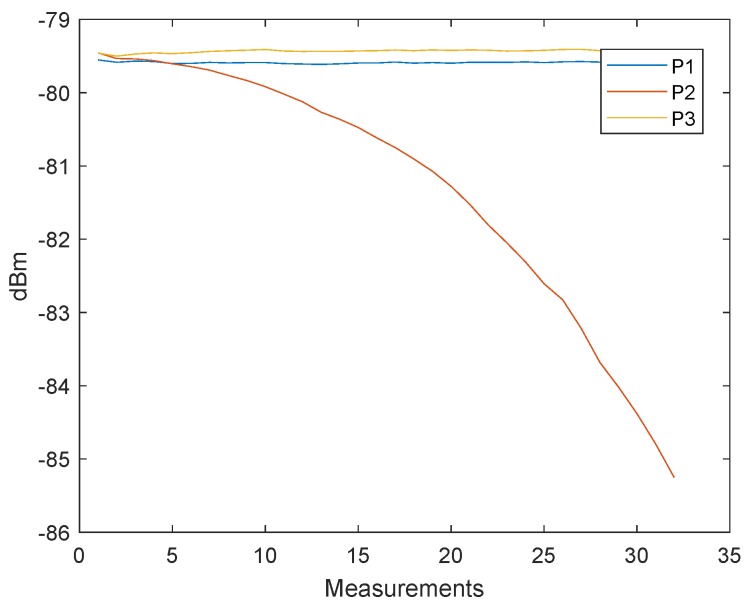
Interference between the received messages P1, P2 and P3. The transmitted signal power of the message P1 and P3 are the same. Due to the short update time are the signals interfering.

**Figure 19 sensors-19-02942-f019:**
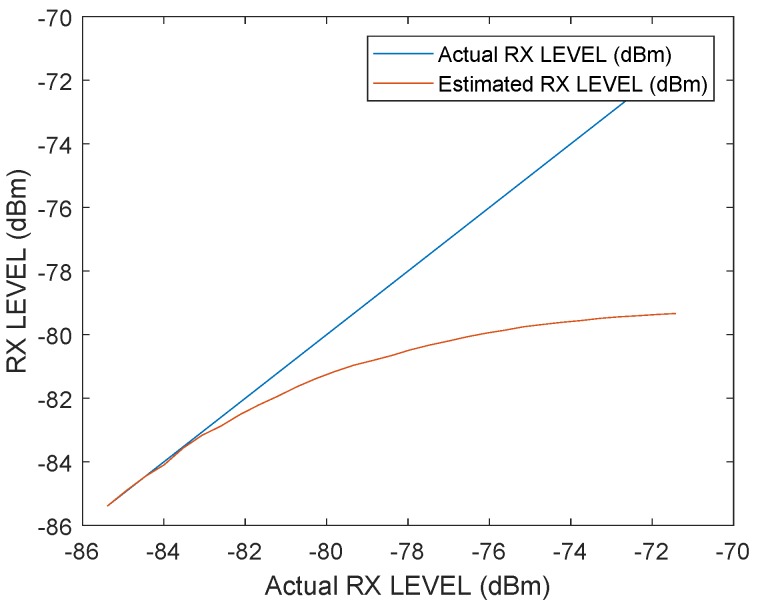
Estimated correction curve between the measured received signal power and the ideal signal power curve.

**Figure 20 sensors-19-02942-f020:**
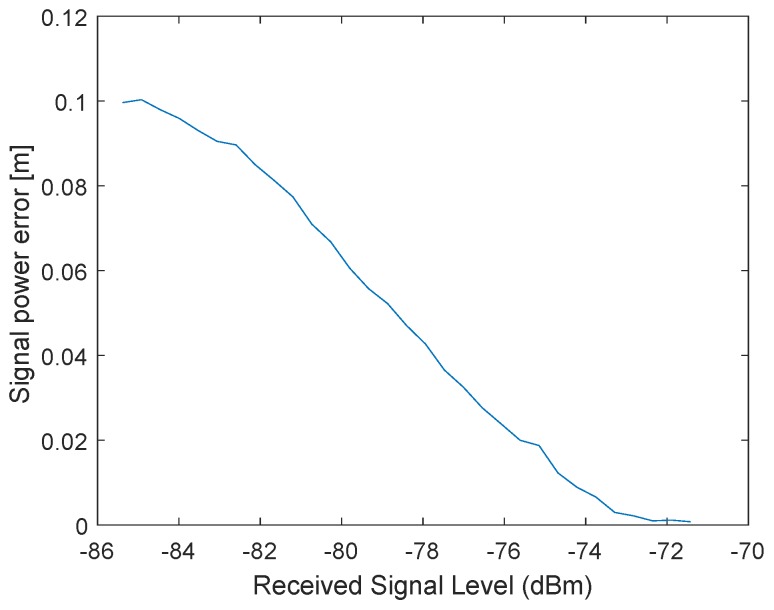
Correction value of the error caused by the signal power as a function of the received signal power.

**Figure 21 sensors-19-02942-f021:**
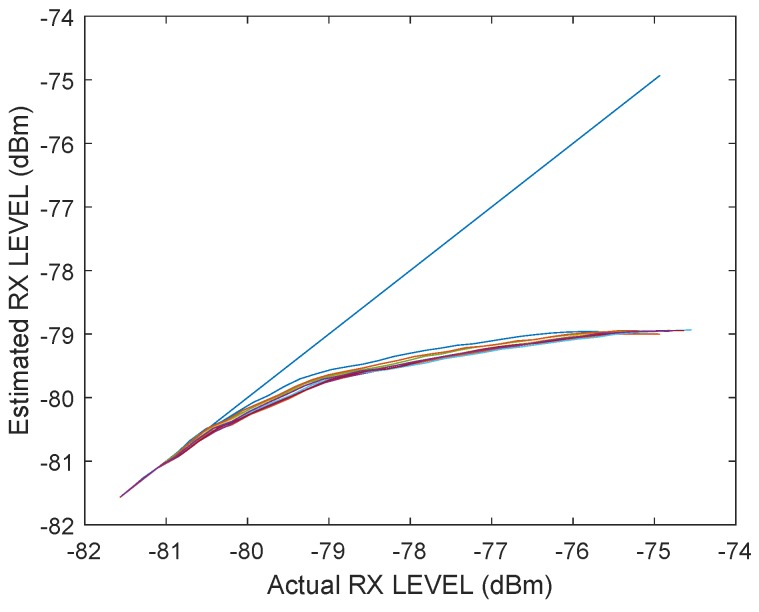
Power correction curve for the received signal power.

**Figure 22 sensors-19-02942-f022:**
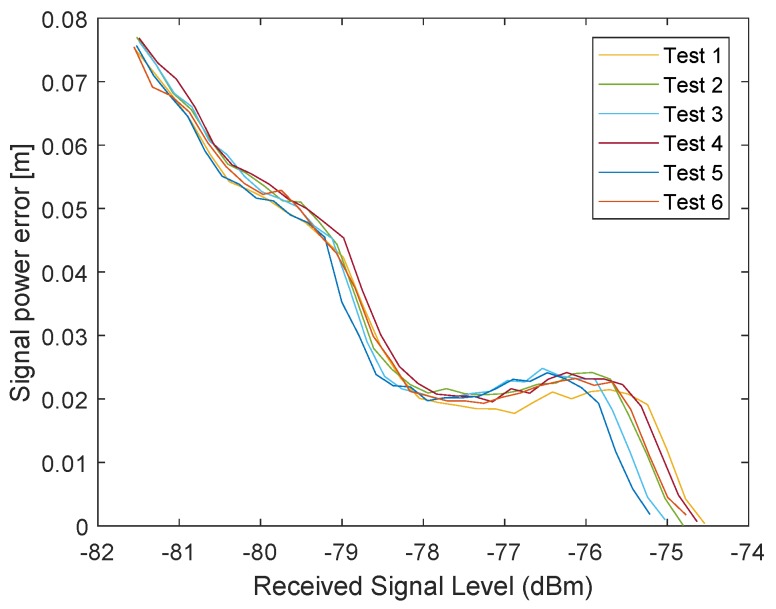
Correction value of the error caused by the signal power as a function of the received signal power for a different station with six restarts.

**Figure 23 sensors-19-02942-f023:**
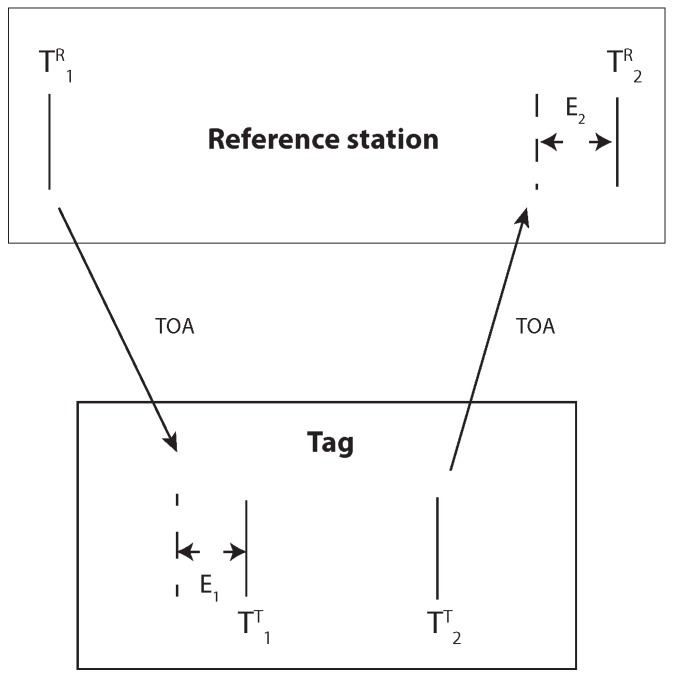
Schematic for the signal power correction for two-way ranging.

**Figure 24 sensors-19-02942-f024:**
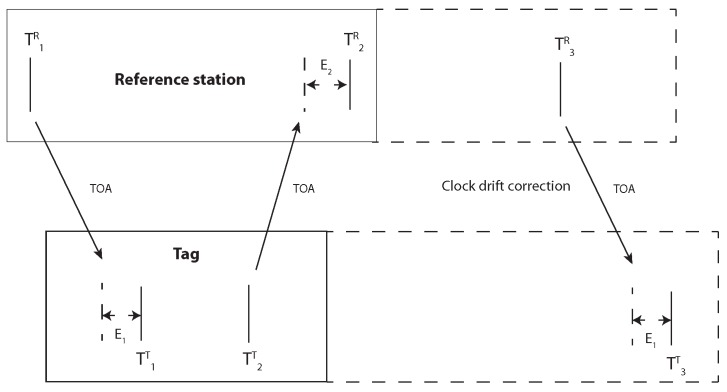
Two-way ranging with clock drift correction between reference station and tag.

**Figure 25 sensors-19-02942-f025:**
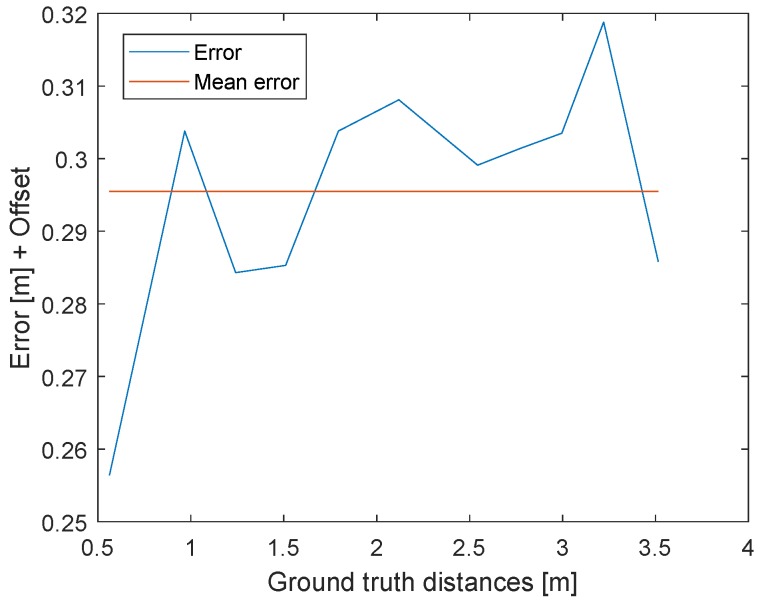
Difference between the measured distances obtained by two-way ranging and the ground truth distances in meter. The constant offset caused by the hardware delay is not compensated but is represented by the red line. The near most constant errors for different distances shows the power correction is correct.

**Table 1 sensors-19-02942-t001:** Test settings.

Parameter	Value
Center Frequency	3993.6 MHz
Bandwidth	499.2 MHz
Pulse repetition frequency	64 MHz
Preamble length	128
Data rate	6.81 Mbps

**Table 2 sensors-19-02942-t002:** Notations used.

Notations	Definition
$T_{i}$	Timestamp
$\Delta T_{n,m}$	Clock drift with respect to the timestamps n and m
$E_{i}$	Timestamp error due to signal power
*Z*	Hardware delay and signal power correction offset

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
