# Peer review of "Decawave UWB Clock Drift Correction and Power Self-Calibration"

_sensors, 2019, doi:10.3390/s19132942_

Round 1

Reviewer 1 Report

The paper seems good but some points can be improved.

In section 2), authors deal with the -41.3dBm/MHz limitation and conclude that the power can be increased for shorter message durations. However, there is a limitation since a peak power of 0dBm must be respected too.

In section 2), I think authors use the word sharper instead of shorter.

Figure 2, 3 and others are not clear. I think authors have to define precisely what is shown on the figures even if this is obvious for them. Titles must be more explicit too. For example, "Integrator of the PLL" is not sufficient to understand what is shown on the figure. Another example, "Signal strength" but what is the considered signal ?

Figure 13 is not clean.

On Figure 25, error bars are shown but I do not understand if they represent rms or peak to peak errors. Moreover, I am not sure that the comment about the 0.3m offset is enough (in the same time, I think "which causes" is more appropriate than "which cause").

Finally, it seems that authors compare their method with laser measurements but do not qunatify the improvement with the method use by Decawave. By doing this, the interest of the proposed work will be highlighted. Moreover, it could be interesting to compare the error obtain with the different UWB ranging methods or others.

Reviewer 2 Report

The paper is adequately written and easy to follow. The paper content is within the journal scope. The proposed approach is interesting, but the presentation needs some improvement. Some remarks are listed in the following.

1) line 43, please change "is UWB" into "UWB is"

2) The jump in Figs 5 and 6 seem to occur when t is about 2300s. If the measurement duration is 50 ms, 20 measurements per seconds occur, and 2300s should correspond to about 46000 measurements, while on line 73 the jump is said to occur after about 4600 measurements. Please verify.

3) line 102: while averaging of 4000 measurements per point is stated when introducing Fig. 10, on page 2 line 67 a 500 point moving average filtering is introduced. Similarly, on line 133 another averaging of 2000 measurements is introduced. Please clarify whether distinct averaging activities are being performed on different quantities, motivating the selected number of samples to be averaged.

4) On line 135, a step size of 3 dB is mentioned, while Fig. 14 shows partial steps of increasing size, where the first step semms to be less than 0.2 dB, the last step seems to be about 0.3 dB, and the overal power variation of signal P2 is less than 1 dB.

Round 2

Reviewer 2 Report

Authors conveniently addressed my remarks